# MARL-LNS: Efficient Multi-agent Deep Reinforcement Learning via Large Neighborhoods Search

## Abstract

Cooperative multi-agent reinforcement learning (MARL) has emerged as a powerful paradigm for addressing complex real-world problems. However, the well-established centralized training with decentralized execution framework is hampered by the curse of dimensionality, leading to prolonged training times and inefficient convergence. In this work, we introduce MARL-LNS, a general training framework that overcomes these challenges by iteratively training on alternating subsets of agents with existing deep MARL algorithms serving as low-level trainers—without incurring any additional trainable parameters. Building on this framework with a specific low-level learner of MAPPO, we propose three variants—Random Large Neighborhood Search (RLNS), Batch Large Neighborhood Search (BLNS), and Adaptive Large Neighborhood Search (ALNS)—each differing in its strategy for alternating agent subsets. Empirical evaluations on both the StarCraft Multi-Agent Challenge and Google Research Football environments demonstrate that our approach can reduce training time by at least 10% while achieving comparable final performance to state-of-the-art methods.

## 1 Introduction

In recent years, multi-agent reinforcement learning (MARL) has split into cooperative multi-agent reinforcement learning and competitive multi-agent reinforcement learning. Competitive MARL has many theoretical guarantees following the previous studies in game theory, and has substantial success in domains like Poker Brown & Sandholm (2018), and Diplomacy Gray et al. (2020). On the other hand, cooperative multi-agent reinforcement learning focuses more on training a group of agents and making them coordinate with each other when everyone shares the same goal, and has also succeeded in many real-world applications like autonomous driving Zhou et al. (2021), swarm control Hüttenrauch et al. (2017), and traffic scheduling Laurent et al. (2021).

Prior work has demonstrated that Multi-Agent Reinforcement Learning (MARL) can converge to robust policies capable of winning games or completing tasks within prescribed time limits. However, efficient training remains a significant challenge due to the dual complexity of optimizing both the reinforcement learning algorithm and the dynamics of multi-agent systems. Historically, the primary bottleneck was the sample efficiency of the RL environments—a predominantly CPU-bound process. This challenge spurred the adoption of parallel sampling strategies and even led to the reimplementation of environments in JAX Rutherford et al. (2024) to harness GPU acceleration. Yet, recent trends indicate that GPUs now account for roughly 60% of the overall training time, shifting the bottleneck to both CPU and GPU resources. This evolution underscores the urgent need for research aimed at accelerating training on both computational fronts.

In this paper, we propose a new learning framework that reduces the time used in the training process of cooperative MARL without harming the performance of the final converged policy. We split the training into multiple iterations, which we called *LNS iterations*, and we only consider the training introduced by a fixed group of agents in each iteration. We choose a subset of agents that are used in training for a certain number of training iterations using existing deep MARL algorithms to update the neural networks, and

then alternate the subsets of agents so that we do not overfit the policy to one certain subgroup of agents while we need to control all of them. We call the group of agents we are considering the *neighborhood*, and call our framework large neighborhood search (MARL-LNS), whose name comes from similar methods used in combinatorial optimization Shaw (1998). Since we have only modified the training at a high level, integrating MARL-LNS with any existing MARL algorithm is both straightforward and simple, and we choose to specifically study how well it performs on the most commonly used MAPPO algorithm in this paper, and show how much speedup our algorithm can provide. Specifically with MAPPO as the low-level algorithm, we provide a theoretical analysis that after multiple LNS iterations, the optimal action learned in this reduced joint action space could still hold the same convergence guarantee provided by MAPPO.

Based on our framework, we provide three simple yet powerful algorithms: random large neighborhood search (RLNS), batch large neighborhood search (BLNS), and adaptive large neighborhood search (ALNS). None of the three proposed have additional parameters that need to be trained, and ALNS do not even have additional hyperparameters that need to be tuned. To show the capability of our framework, our algorithms rely on random choices of which group of agents is used in training and do not include any hand-crafted or learned heuristic to select the neighborhood.

We test our algorithms in many scenarios in the popular StarCraft Multi-Agent Challenge (SMAC) Samvelyan et al. (2019) and Google Research Football (GRF) Kurach et al. (2020) environments, and show that our simple algorithms can reach at least the same level of performance while we can be around 10% faster than the baselines in terms of total wall clock time. We provide an ablation study on how the number of agents in the training neighborhood at each period, i.e., the neighborhood size, affects the performance of the algorithms and the training time.

## 2 Related Works

Cooperative multi-agent reinforcement learning has been a popular research topic in the past few years. The most popular solution framework is the centralized training decentralized execution (CTDE) framework, in which two of the most popular lines of research are the value-based research like VDN Sunehag et al. (2017), QMIX Rashid et al. (2018), QTRAN Son et al. (2019), and policy-based research like MADDPG Lowe et al. (2017), COMA Foerster et al. (2018) and MAPPO Yu et al. (2022). A few recent works propose using only local neighborhood information at each timestep to keep the joint state-action space small Zhang et al. (2022). Our algorithm framework is different from those by not changing the neighborhood used at each timestep in one episode, but only changing it after enough training iterations. Other works like VAST Phan et al. (2021) proposed to factorize the coordination into a sum over a few smaller sub-groups with smaller joint-action space, while our algorithm will only train one sub-group at one time. Iqbal et al. (2021) also proposed to factor the agents into random subpartitions in value-based MARL, and our work is different from theirs in training in more than two groups and working on policy-based MARL. DIFFER Hu et al. (2023) also proposed to use part of the data in training, but they focused on training the policy efficiently in the beginning, while we are focusing on the overall efficiency and effectiveness.

The idea of large neighborhood search has been extensively used in combinatorial optimization problems since proposed Shaw (1998), such as vehicle routing Ropke & Pisinger (2006) and solving mixed integer linear programming Munguía et al. (2018); Song et al. (2020). Similar ideas have also been used to help convergence in game theory by fixing a group of agents Nair et al. (2003). Recently, the same technique has been introduced to multi-agent path finding, where they fix the path of a group of agents and replan the path of other agents at each iteration Li et al. (2021); Huang et al. (2022); Li et al. (2022); Wang et al. (2024). In these algorithms, the decisions of the variables/agents in the chosen neighborhood are updated such that this "move" results in a better overall solution to the optimization problem. In this paper, we propose a framework that introduces similar ideas into the MARL community. In one special case of our algorithm, our algorithm becomes iterative learning, which has been widely used in equilibrium findings Daskalakis et al. (2010); Wang et al. (2018); Chen et al. (2021); Nair et al. (2003). While their works mostly use iterations to make the training more stable and smooth, our work also focuses on reducing the time in each training iteration.

---

**Algorithm 1** MARL-LNS: Large neighborhood search (LNS) framework used in this paper with MAPPO as low-level algorithm.

---

1: Initialize value network $V$ and policy network $\pi$.
2: **repeat**
3:     Choose the neighborhood $G = NeighborhoodSelect()$.
4:     **repeat**
5:         Reset the replay buffer
6:         **repeat**
7:             Sample trajectories $\tau = (\tau_1, \tau_2, \tau_3, \ldots, \tau_n)$ from the environment according to $\pi$
8:             Save $\tau_{g_1}, \tau_{g_2}, \ldots, \tau_{g_m}$ to the replay buffer, where $G = (g_1, g_2, \ldots, g_m)$
9:         **until** Sampled $Buffer\_length$ trajectories
10:         Train $V$ and $\pi$ with the replay buffer using MAPPO
11:     **until** Trained $N_{Training\_per\_neighborhood}$ rounds.
12: **until** Has done $N_{LNS\_iterations}$ Iterations

---

While earlier works mostly focus on improving the speed in sampling on the CPU side Rutherford et al. (2024), and making parallel sampling a common practice, recently, there has been a growing interest in optimizing the overall training time, which takes GPU time into account. Yu et al. (2023) proposed an asynchronous MARL based on MAPPO, thus reducing 10.07% actual training time compared to MAPPO. Gogineni et al. (2023) explored optimizing based on the locality of a sampling strategy to improve cache locality and reached a 10.2% end-to-end training time reduction. Chang et al. (2022) proposed to guide agents with programs designed for parallelization, outperform the baselines in completion rate, and reduce the overall time by 13.43%. Because the actual time reduction heavily relies on the specific configuration of the CPU and GPU used in training, the time reduction numbers are not comparable across different papers. Furthermore, these works that try to reduce the overall training time are orthogonal to each other, and ideally, all the methods, including the one that we will propose here, can be combined together to get a huge overall reduction.

## 3 Preliminaries

In multi-agent reinforcement learning, agents interact with the environment, which is formulated as a decentralized partially observable Markov decision process (Dec-POMDP). Formally, a Dec-POMDP is defined by a 7-tuple $\langle S, A, O, R, P, n, \gamma \rangle$. Here, $n$ is the total number of agents, $S$ is the state space, and $A$ is the action space. $O$ is the observation function. $P$ is the transition function. $R(s, a)$ denotes the shared reward function that every agent received. $\gamma$ is the discount factor used in calculating the cumulative reward. During the episode, agent $i$ use a policy $\pi_i(a_i|o_i)$ to produce an action $a_i$ from the local observation $o_i$. The environment starts with an initial state $s^1$, and stops when the environment provides a true in the *done* signal. The objective is to find a policy $\pi = (\pi_1, \pi_2, \ldots, \pi_n)$ within the valid policy space $\Pi$ that optimizes the discounted cumulative reward $\mathcal{J} = \sum_t \mathbb{E}_{s^t, a^t \sim \pi} \gamma^t \cdot r^t$, where $s^t$ is the state at timestep $t$, $r^t = R(s^t, a^t)$, and $a^t$ is the joint action at timestep $t$. A trajectory $\tau$ is a list of elements that contains all the information of what happens from the start to the end of one run of the environment, $\tau = <s^1, a^1, r^1, s^2, a^2, r^2, \ldots, s^t, a^t, r^t>$. During the training of CTDE algorithms, the trajectories are split into trajectories in the view of every agent $\tau = (\tau_1, \tau_2, \ldots, \tau_n)$, which contains useful information in terms of each agent and forms an individual trajectory. The information included in each $\tau_i$ depends on the algorithm that is used. For example, MADDPG needs to train a centralized value function, $\tau_i$ is the same as $\tau$. However, for MAPPO with GAE, the value function only needs the centralized state but not the joint action, so $\tau_i = <s^1, a_i^1, r^1, \ldots, s^t, a_i^t, r^t>$ which reduce the joint action as needed.

# 4 Large Neighborhood Search for MARL

## 4.1 Large Neighborhood Search Framework

Large Neighborhood Search (LNS) is a widely adopted meta-heuristic in combinatorial optimization and multi-agent pathfinding, aimed at efficiently finding high-quality solutions for problems where obtaining the optimal solution is computationally prohibitive. The method begins with an initial solution and selects a subset—referred to as a *neighborhood*—which is then partially *destroyed*. The optimizer subsequently *rebuild* the solution for the destroyed components while keeping the remainder fixed, thereby enhancing computational efficiency by reducing the solution space during each optimization iteration.

While the high-level idea can lead to many completely different algorithms, we now present a detailed description of our framework within MARL. During training, we maintain a fixed group of $m$ agents—referred to as a *neighborhood* $G = \{g_1, g_2, \ldots, g_m\}$, where $m$ is a hyperparameter determining the neighborhood size. Next, we sample a batch of data from the environment. In this process, the complete trajectory $\tau$ is partitioned into $n$ subtrajectories, $(\tau_1, \tau_2, \ldots, \tau_n)$, where $\tau_i$ designated for training agent $i$. We then retain only the subtrajectories corresponding to the agents in the neighborhood—i.e., we store $\{\tau_{g_1}, \tau_{g_2}, \ldots, \tau_{g_m}\}$ in the replay buffer rather than the entire trajectory $\tau$. Once sufficient data for a training batch has been collected, we perform a training update using established algorithms such as MAPPO and subsequently clear the replay buffer. This approach ensures that the training data is exclusively focused on the agents within the neighborhood. We repeat the trajectory sampling process multiple times until a new group of agents is selected as the neighborhood. Each complete cycle of sampling trajectories and training for a fixed neighborhood is defined as an *LNS iteration*. We continue with several LNS iterations until we meet our preset stopping criterion—typically defined by a total number of steps sampled—consistent with standard practices in MARL.

In contrast to traditional LNS approaches in other domains, our MARL framework preserves the existing policies of agents within the neighborhood rather than *destroying* them. This decision is motivated by the substantial computational cost of retraining policies from scratch. Consequently, the *rebuild* phase in our approach turns into continuous optimization of the current policies using trajectories from the agents in the neighborhood. Suppose the neighoborhood size $m$ is to be fixed in the process and the average CPU-GPU wallclock time ratio is $\kappa = \frac{T_{CPU}}{T_{GPU}}$, the theoretical time saved ratio $\eta$ will satisfy

$$\eta = \frac{n + m\kappa}{m + m\kappa}. \tag{1}$$

Unlike many existing works focusing on sample efficiency for MARL Rutherford et al. (2024), our algorithms gain efficiency by using fewer data for backpropagation in each batch of training. As a high-level approach, our algorithm is agnostic to how the data is used for training and how coordination is solved in the framework, i.e., how loss and gradient will not be changed at all as long as they are using the data we provided in the replay buffer. Because of this, the lower-level MARL algorithm, which uses the data to train the neural network, is very flexible. In this paper, we choose to use MAPPO Yu et al. (2022) as the low-level algorithm as an example. In MAPPO, a centralized value function concatenates all observations from all agents in the environment. Even if some agents are not included in the neighborhood and thus not used in training, their observations are still kept in the input of the value function as global state information. Besides, the trajectory of each agent contains exactly the same state information as the original trajectories, and removing the trajectories outside the neighborhood saves us the space of copying the action information of those agents. We provide this algorithm framework in Alg. 1, where lines 3, 8, 11, and 12 are the lines that are introduced because of our LNS framework.

Next, we theoretically show that the convergence guarantee will not be affected by the introduced framework, i.e., as long as the low-level algorithm can learn to cooperate, our framework will still be able to learn to do the same. From intuition, our algorithm can be viewed as an extension of iterative training that has been commonly used in MARL to stabilize the training and even improve the final policy, especially in competitive MARL Daskalakis et al. (2010); Wang et al. (2018); Chen et al. (2021). More formally, our

algorithm can be reduced to a block coordinate descent algorithm (BCD) by letting each variable used in the optimization be the policy of each agent, and the objective value is our reward function. BCD is studied a lot by optimization theory researchers Tseng (2001); Beck & Tetruashvili (2013); Lu & Xiao (2015), and its convergence rate is proved under different conditions. Here, we provide a convergence guarantee that specifically proves that with MAPPO as the low-level algorithm, the expected cumulative reward of the learned policy from MARL-LNS is the same as the one from MAPPO:

**Theorem 1** *(Adapted from Lyu & Li (2020)) Assume the expected cumulative reward function $\mathcal{J}$ is continuously differentiable with Lipschitz gradient and convex in each neighborhood partition, and the training by the low-level algorithm guarantees that the training happening on the i-th neighborhood is bounded by a high-dimension vector $w_i$ on the joint policy space $\Pi$. Define the optimality gap as $\Delta_i(\pi) := \sup_{\hat{\pi} \in \Pi, |\hat{\pi} - \pi| \leq c' w_i} J_{\hat{\pi}} - J_{\pi}$, where $c'$ is a constant. Suppose $\sum_{i=1}^{\infty} w_i^2 < \infty$, and let the policy after the k-th LNS iteration be $\pi^k$. If the optimality gap is uniformly summable, i.e., $\sum_{i=1}^{\infty} \Delta_i < \infty$, then there exists some constant $c > 0$ such that for $i \geq 1$,*

$$min_{1 \leq k \leq i} \sup_{\pi_0 \in \Pi} [- \inf_{\pi \in \Pi} \langle \nabla \mathcal{J}_{\pi^k}, \frac{\pi - \pi^k}{|\pi - \pi^k|} \rangle] \leq \frac{c}{\sum_{k=1}^{i} w_k} \tag{2}$$

**Proof Sketch** This is a direct application of Theorem 1 in Lyu & Li (2020), wherein we reformulate the MARL optimization problem as a block coordinate descent problem. In this formulation, each agent's policy is treated as an optimization variable, while the cumulative reward function—derived from these policies—serves as the objective function.

Specifically, the assumptions on $w_i$ are common assumptions for convergence in MARL, and are usually handled by the learning rate decay mechanism in the learning optimizer together with the clip mechanism in reinforcement learning algorithms like TRPO and PPO. Furthermore, because the expected reward function $\mathcal{J}$ is based on policy rather than action, the continuously differentiable condition is also satisfied in environments with a continuous policy space. This theorem guarantees that the convergence of MARL-LNS is irrelevant to the neighborhood size $m$ as well as what is included in each neighborhood. However, the learned policy could still be empirically worse than the policy learned by the low-level algorithm if the learning rate is not handled properly.

### 4.2 Random Large Neighborhood Search

---
**Algorithm 2** Neighborhoood selection function for Adaptive Large Neighborhood Search (ALNS).

---
1: Initialize neighborhood size $m = 2$.
2: **function** NEIGHBORHOODSELECT
3:     **if** Agent performance was not improving in the last two LNS iterations **then**
4:         $m = min(m + 2^{\lfloor log_2^m \rfloor - 1}, \lceil \frac{n}{2} \rceil)$
5:     **end if**
6:     $R = random.choice(n, m)$
7:     **return** R
8: **end function**

---

While many previous works of large neighborhood search in CO and MAPF focus a lot on neighborhood selection, in this paper, we show the capability of our framework by choosing the neighborhood randomly and do not introduce any hand-crafted heuristics. We leave some discussion on how some simple heuristic-based approaches to neighborhood selection do not help the framework learn a better policy more efficiently in the appendix. Specifically, the neighborhood selection part is instantiated with uniformly sample m agents from 1,..,n without replacement. We call this algorithm the random large neighborhood search (RLNS).

### 4.3 Batch Large Neighborhood Search

While pure random can introduce a lot of variance to the training, here we also provide an alternative batch-based large neighborhood search (BLNS) algorithm, which differs from RLNS in the neighborhood

selection function. Unlike RLNS, before any training starts, we create one permutation $(p_1, p_2, \ldots, p_n)$ of all agents. Again, for simplicity, the permutation is created randomly in this paper. After creating the permutation, we select the agents in order whenever we want to select the next group of neighborhoods. In other word, given a fixed neighborhood size $m$, the first neighborhood would be $\{p_1, p_2, \ldots, p_m\}$, the second would be $\{p_{m+1}, p_{m+2}, \ldots, p_{2m}\}$, and keep going like this. If $m$ cannot be divided by $n$, the neighborhood that includes the last agent $p_n$ will also include agent $p_1$, and keep going from $p_2, p_3$ to $p_n$ again.

### 4.4 Adaptive Large Neighborhood Search

While the RLNS and BLNS use a fixed neighborhood size, the adaptive large neighborhood size is recently becoming popular in the large neighborhood size community in combinatorial optimization and multi-agent path finding Sonnerat et al. (2021); Huang et al. (2022). Here, we propose another variant of MARL-LNS that adaptively changes the neighborhood size. In the beginning, we define a list of $k$ potential neighborhood size $M = [m_1, m_2, \ldots, m_k]$, where $m_1 < m_2 < \cdots < m_k$. In training, if in the last two LNS iterations, the evaluation performance is not getting any improvement, the current neighborhood size $m_i$ will be changed to $m_{i+1}$ in the next LNS iteration unless $m_i = m_k$ already. While this is orthogonal to the previously mentioned RLNS and BLNS, which focus on neighborhood selection, we combine this method with RLNS as a new algorithm, the adaptive large neighborhood search (ALNS). For implementations, because most environments have a smooth reward that encourages more agents to collaborate together, we can stay on a small neighborhood size most of the time. We advocate for setting $m_1 = 2$ and $m_i = \min(m_{i-1} + 2^{\lfloor \log_2(m_{i-1}) \rfloor - 1}, \lceil \frac{n}{2} \rceil)$, where $n$ is the total number of agents, to ensure a gradual increase in neighborhood size, optimizing both efficiency and effectiveness. The corresponding ALNS pseudo-code is presented in Alg. 2.

Besides the original benefit from MARL-LNS, the gradually growing neighborhood size $m$ gives an additional benefit that fits the nature of MARL: At the beginning of the training, both the value network and the policy network are far from accurate and optimal. In this period, MARL algorithms are mostly training value functions, and the neighborhood size does not affect the training of the value function. Later on, training on a subset of agents makes the training similar to iterative training, which reduces the size of the joint action space to speed up the convergence to local optimums. When it comes to the end of the training process, the neighborhood size will become large enough to cover the need of many agents in the environment to collaborate on a single task.

## 5 Experiments

### 5.1 Experimental Settings

In this paper, we test our results on both StarCraft Multi-Agent Challenge (SMAC) and Google Research Football (GRF) environments. We use parameter sharing between agents because it has been shown to improve the training efficiency while not harming the performance Christianos et al. (2021). We use some common practice tricks in MAPPO, including Generalized Advantage Estimation (GAE) Schulman et al. (2015) with advantage normalization, value clipping, and including both local-agent specific features and global features in the value function Yu et al. (2022). For our algorithms, instead of setting a number of how many times of training is used for each LNS iteration, we use an equivalent version of providing the total number of different neighbors $N_{Training\_per\_neighborhood} = N_{LNS\_iterations}/N_T$, where $N_T$ is the new hyperparameter we control. By default, the neighborhood size is half of the total number of agents, as we will later show in the ablation study that this default setting will keep the policy to no worse after training in all our experiments while providing a reasonable time reduction. To show that our algorithm does not introduce extra fine-tuning effort, we do not change the hyperparameters used in our low-level algorithm MAPPO, e.g., the learning rate, the batch size, etc, as well as the network designs, and environment configurations. We provide more details in the appendix. This will affect the conclusion of Thm. 1, but we will use our results to show that this is not affecting the effectiveness of our algorithm. For ALNS, we use the candidate neighborhood size list as we recommended.

## 5.2 SMAC Testbed

We test our algorithms on 5 different random seeds. For each random seed, we evaluate our results following previous works: we compute the win rate over 32 evaluation games after each training iteration and take the median of the final ten evaluation win rates as the performance to alleviate the marginal distribution in this environment. We compare our algorithm with MAPPO, IPPO, and QMIX. We only test our results in scenarios classified as hard or super-hard, since many easy scenarios have been perfectly solved, and our algorithm will become iterative training, which has been studied a lot, in scenarios that include only 2 or 3 agents.

We report our results in Table. 1 and Table. 2. In Table. 1, we observe that the time reductions are consistent across scenarios since the reduction comes from reducing the training data used. The time reduction is greater than that of other previous works on speeding up the overall time used in the training of MAPPO. Besides, comparing the reduction between the full training and the early number of steps results, we found that most savings are from the first half of the training, where the neighborhood size stays at a very small value, and the MARL algorithm is getting improvement on both the value network and the policy network given the huge space for improvement in this phase. In Table. 2, the win rate of our algorithms is at least as good as the current algorithms while actually getting a higher final win rate in difficult scenarios like 5mvs6m and MMM2. This shows that our algorithm does not actually trade effectiveness in the trained policy for training efficiency, but gets the speedup without harming the performance. Furthermore, RLNS generally has a bigger variance in win rate than BLNS, which is coming from the pure-random-based neighborhood selection, but this randomness also enables RLNS to get a higher median win rate in the very hard 3s5zvs3s6z scenario where BLNS fails in 3 seeds and ends up with a low median value. When the neighborhood does not include both types of allies in 3s5zvs3s6z, the evaluation win rate drops quickly and needs a lot of extra training effort to make up for this drop. On the other hand, this scenario itself usually has a big marginal distribution, and both MAPPO, RLNS, and ALNS are still getting a policy with a win rate of less than 30% in 2 out of the 5 seeds, leaving a great space for more stable policy training. Our ALNS is always one of the best algorithms given that it will at last use half of the total number of agents, but on the other hand, it never outperforms other algorithms, mostly because they are still in the same high-level framework.

|          | RLNS & BLNS | ALNS | ALNS (50% Total Steps) | ALNS (70% Total Steps) |
|----------|-------------|------|------------------------|------------------------|
| 5mvs6m   | 5%          | 5%   | 5%                     | 5%                     |
| MMM2     | 21%         | 18%  | 21%                    | 19%                    |
| 3s5zvs3s6z | 8%        | 12%  | 15%                    | 14%                    |
| 27mvs30m | 10%         | 16%  | 23%                    | 20%                    |
| 10mvs11m | 12%         | 19%  | 22%                    | 21%                    |

Table 1: Average total time reduction used by MARL-LNS compared to MAPPO in SMAC. The 50% and 70% Total Steps indicate the scenarios in which each algorithm completes a respective portion of the total number of steps.

|          | MAPPO       | IPPO        | QMix        | RLNS (ours) | BLNS (ours) | ALNS (ours) |
|----------|-------------|-------------|-------------|-------------|-------------|-------------|
| 5mvs6m   | 89.1 (2.5)  | 87.5 (2.3)  | 75.8 (3.7)  | **96.9 (8.2)** | **96.9 (3.6)** | **96.9 (3.6)** |
| MMM2     | 90.6 (2.8)  | 86.7 (7.3)  | 87.5 (2.6)  | **96.9 (31.8)** | **96.9 (4.7)** | 93.9 (2.4)  |
| 3s5zvs3s6z | 84.4 (34.0) | 82.8 (19.1) | 82.8 (5.3)  | **87.5 (44.0)** | 12.6 (31.8) | **87.5 (34.7)** |
| 27mvs30m | **93.8 (2.4)** | 69.5 (11.8) | 39.1 (9.8)  | 90.6 (2.5)  | **93.8 (7.2)** | **93.8 (4.7)** |
| 10mvs11m | **96.9 (4.8)** | 93.0 (7.4)  | 95.3 (1.0)  | 93.8 (5.3)  | **96.9 (2.4)** | **96.9 (2.4)** |

Table 2: Median evaluation win rate and standard deviation on SMAC testbed.

### 5.2.1 Ablation Study on Neighborhood Size

After showing that our algorithms are good in terms of learning the policies, we now use an ablation study on neighborhood size to show how our method provides flexibility to trade off a tiny win rate for faster training

time. As a special case, when changing the neighborhood size to 1, BLNS is iterative training, and when the neighborhood size is as big as the total number of agents, our framework is the same as the low-level algorithm MAPPO. We did not change the total environment steps because we do not see any benefit in doing that.

We test BLNS on the 27mvs30m scenario, because it has the biggest number of agents. We show our results in Table. 3. We observe that when increasing the neighborhood size, the final win rate is improving, while the time usage is also bigger. When we set the neighborhood size $m$ to 10, the final performance is within one standard deviation of the low-level algorithm, while the training is 15% faster. And if we set the neighborhood size $m$ to 5, the final performance is within two standard deviations of MAPPO, with only 77% of the original training time used. These numbers almost perfectly match our theoretical time used saving shown in Eq. 1. ALNS also achieves an average time saving between a neighborhood size of 5 and 10 and a final win rate as good as MAPPO, showing that it is a reasonably good algorithm that balances the performance of learned policy and training speed. Overall, ALNS and BLNS with $m = 5$ are the two most dominant settings on the Pareto frontier of total training time and final win rate. All the savings are because the sampling time in CPUs is only taking 44% of the total training time of MAPPO, as shown in Table 3. All other time is spent on transferring data between CPU and GPU and updating the neural networks on GPU, which is the time related to updating that can be largely saved by removing part of the training data. On the other hand, we can also observe that the standard deviation of the win rate of the policies from a small neighborhood size, i.e., $m = 1$ or $m = 3$, is growing bigger by the end of the training. This is because their policies have not actually converged, and given long enough time, their performance could also reach a good result. But understanding that the primary focus is training efficiency, they are not allowed to train any longer.

| m | 1 | 3 | 5 | 10 | 15 | 27 (MAPPO) | ALNS |
|---|---|---|---|---|---|---|---|
| Win Rate (%) | 3.1 (3.9) | 62.5 (28.3) | 87.5 (5.3) | 90.6 (8.2) | 93.8 (7.2) | 93.8 (3.8) | 93.8 (4.7) |
| Training Time (s) | 7.14 | 7.35 | 7.37 | 8.22 | 9.29 | 9.52 | 8.09 |
| Updating Time (s) | 2.10 | 2.31 | 2.35 | 4.02 | 5.10 | 5.33 | 3.88 |

Table 3: Median value and the standard deviation on evaluation win rate for MARL-RLNS with varied neighborhood sizes $m$, together with their average training time and average updating time (training time includes both sampling time and updating time with a little overlap) for 1k episodes on 27m_vs_30m scenario from SMAC. Specifically, when $m = 27$, RLNS is the same as MAPPO.

### 5.3 GRF Testbed

We evaluate our results following the common practice in GRF: compared to the SMAC above, instead of evaluating in 32 evaluation games, the policies are evaluated in 100 rollouts, and instead of reporting the median value, the mean value is reported. Because most scenarios in GRF have less than 6 agents, we only test the algorithm in the corner scenario with 5 different random seeds.

Our results are shown in Table. 4. Even if this test case is naturally heterogeneous, we observe that giving a good hyperparameter to BLNS and RLNS will give our algorithm the same level of performance as MAPPO that is within one standard deviation, and applying the same group of hyperparameters to RLNS can learn a slightly worse policy with larger variance. In this hyperparameters setting, BLNS and RLNS are trained at least 14% faster than MAPPO while ALNS is 25% faster than MAPPO. Additionally, GRF shows a more significant difference than SMAC when changing the number of different neighborhoods used in the training process. Medium size of 20 is enough for algorithms to explore collaborations with other agents while not changing it so regularly and introducing instability to the training. Again, the overall time reduction is extremely close to the theoretical time-saving in Eq. 1. This demonstrates that our algorithm can predictably provide a time reduction as we expected.

| | Win Rate (%) | Time Reduction (%) |
|---|---|---|
| BLNS ($m = 7, N_T = 20$) | 65.6(8.1) | 15 |
| BLNS($m = 5, N_T = 20$) | 57.4(6.0) | 22 |
| BLNS($m = 7, N_T = 10$) | 42.4(3.1) | 15 |
| BLNS($m = 7, N_T = 5$) | 42.2(1.8) | 14 |
| BLNS($m = 7, N_T = 40$) | 50.4(3.1) | 15 |
| RLNS($m = 7, N_T = 20$) | 58.0(13.5) | 15 |
| ALNS($N_T = 20$) | 63.0(14.7) | 25 |
| MAPPO | 65.53(2.19) | 0 |

Table 4: The average evaluation win rate of BLNS in different settings compared to RLNS and other baselines on GRF, together with their corresponding time reduction compared to MAPPO.

## 6 Discussions

### 6.1 Limitations

While our algorithms are performing well in SMAC and GRF, there are two major limitations. First, our algorithmic framework MAPF-LNS relies on the fact that in big and complex environments, not all agents would be used to fulfill a single task, and agents would learn to split the job into sub-tasks and do their own parts. However, the Multi-domain Gaussian Squeeze (MGS) environment used in the Qtran paper Son et al. (2019) is one counter-example that requires strong cooperation within all agents and thus leads to unstable and slow convergence when using our algorithms. Second, our current random-based neighborhood selection algorithms assume that all agents are similar, and no agents have a higher priority, higher importance, or significant differences from others in the environment. This is not applicable in some real-world scenarios where certain agents should have higher priority, for example, a fire engine in the traffic system. For the same reason, the marginal distribution could be ignored during training. This may lead to fairness issues if the neighborhood selection is not taking these into account (as in the proposed RLNS and BLNS), and this could lead to a potential negative social impact.

### 6.2 Naming

About the naming of our algorithm, we have considered using a different name for our algorithm that does not include LNS. However, two primary reasons led to our decision to keep the name with LNS: 1). Although our algorithm is not fully aligned with other LNS approaches, our algorithm is still optimizing on a local neighborhood at each iteration, which is the essence of an LNS algorithm. While the high-level idea of training on subsets of agents has been explored and called differently in different communities, e.g., subset-based optimization, random coordinate descent, and LNS, using the name of LNS could connect our future work with the rich literature in the combinatorial search community on LNS, for example, neighborhood selection. 2). We believe that our algorithm is the closest algorithm to LNS in the MARL community. The main difference between our framework and a typical LNS algorithm, which usually only changes the selected neighborhood subset, is that our MARL RLNS and BLNS do not completely fix the policy of agents that are not in the neighborhood because of parameter sharing between the policy of different agents. However, the common practice of using parameter sharing in MARL is the key to making the training efficient. Our improvement in terms of time is slightly smaller than the one brought by parameter sharing, so we believe it is necessary to keep parameter sharing in the framework.

### 6.3 Theorem on Convergence

Remark that in this paper, we are addressing the efficiency while not expecting the algorithm to outperform the low-level algorithms, we do not guarantee the number of training per LNS neighborhood is long enough for the value function and policy function to converge. Additionally, in the most recent iteration of the paper, another update of the theorem is introduced, which, diverging from previous requirements, does not necessitate the cumulative reward function to be Lipschitz differentiable within each neighborhood partition, instead necessitating a weaker inequality condition. Given that differentiable is common in the proof of

convergence in numerous other Multi-Agent Reinforcement Learning (MARL) algorithms, which are highly likely to be employed as the low-level algorithm to guarantee the algorithms can learn to cooperate, we have elected not to incorporate the update here.

### 6.4 Neighborhood Selection

While neighborhood selection is one of the most important parts of LNS research, in this paper, our neighborhood selection strategy is purely random based. There are three major reasons for this: 1. Random selection itself is very strong and robust in domains that are not well-studied by LNS researchers. Many proposed heuristics for neighborhood selection failed to outperform random selection on domains that are not included in their paper or even much worse. While the primary purpose of this paper is to introduce LNS into the context of MARL, a pure random-based algorithm variant that stably outperforms the low-level MAPPO is already good enough to fulfill the objective. 2. While MARL is a very general algorithm that could be used for many environments, proposing a very strong and general neighborhood selection algorithm is a very big challenge. If one wants to bring the generalizability of machine learning into this domain, one might need to think about how to tradeoff the additional time used in training, such as a neighborhood selection model, and also the inference time used in the training process may reduce the total time reduction. 3. Even in the case that we do not want generalizable heuristics, we have tried in SMAC based on geographical clustering to choose local neighborhoods to align with the previous works in geographical clustering Zhang et al. (2022). This works no better than random, and is particularly bad in the 3s5zvs3s6z scenario, whereas we discussed in the main paper, including both type of agents are necessary for each neighborhood to get a strong policy. 4. Although there are only two main branches in cooperative MARL, namely value-based and policy-based algorithms, the concrete training details are completely different from one MARL algorithm to another. And what is available for a neighborhood selection algorithm to use is also very different. For example, for MAPPO, the algorithm only has a joint value function learned, and the optimization is based on the advantage function in each episode, while in QMIX, each agent also has a local Q function that takes action into account. These differences can make a neighborhood selection algorithm like choosing the agents whose local Q function is the smallest in Qmix not applicable to MAPPO. But our current random-based algorithm does not have such a problem, so we really recommend the current group of RLNS, BLNS, and ALNS as a general solution if one just wants to get an easy speedup, no matter what their low-level algorithm is.

### 6.5 Rejecting Bad Neighborhoods

In the current version of the paper, we do not reject any bad neighborhoods, primarily because the training of MARL always includes a lot of fluctuations caused by both local optimal and exploration, as shown in Fig. 1 and Fig. 2. If we reject a neighborhood just by ad-hoc reduction of the evaluation win rate, the learned policy will end up in an early-point local optimal that is far from optimal. Another minor reason is that our main focus of this paper is to provide this robust MARL-LNS framework that gains the total training time reduction with nearly no extra effort, and clever rejection criteria are something that one may see as a huge burden and are what we want to avoid. However, it is undeniable that developing a clever rejecting heuristic can help the algorithms to be more stable, and we would like to leave such research in the future.

### 6.6 Improvement Margin

In this paper, our improvement is from 5% to 25% , depending on the scenarios and settings of our algorithms. It needs to be addressed that our reported results are obtained on a server that highly prioritizes the capacity of GPU, and thus the sampling time (which is CPU dependent) takes a larger portion of time than it would on a server that balances the configuration of CPU and the GPU. When moving to such a server that replaces the V100 with an NVIDIA P100 GPU, the improvement for 5mvs6m scenario in SMAC, which is the scenario that we have the least improvement ratio, increases from 5% to 15%, and could reach a time saving of 25% when everything is done on a single 16-core xeon-6130 CPU with 32GB memory without any GPU.

## 7 Conclusion

In this paper, we introduce MARL-LNS, a novel large neighborhood search framework for cooperative multi-agent reinforcement learning that enhances training efficiency by leveraging subgroups of agents—termed neighborhoods—during each training iteration. Building on this framework, we develop three distinct algorithms—RLNS, BLNS, and ALNS—each employing a unique strategy for neighborhood selection. Importantly, our approach introduces no additional trainable parameters, with ALNS being entirely hyperparameter-free. Through rigorous theoretical analysis and comprehensive empirical evaluations, we demonstrate that our methods significantly improve training efficiency without compromising any critical aspects of performance. This efficiency gain is particularly pronounced in challenging environments such as MMM2 within SMAC, where our algorithms also facilitate the discovery of superior policies.

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

## A   Implementation Details for Experiments

Here we provide the hyperparameter table for our experiments. While most results in the experiment section are from previous papers Yu et al. (2022) and Wu et al. (2021), we only provide the hyperparameter for our algorithms.

Table. 5 provides the neighborhood size used specifically for each scenario, while Table. 6 provide other hyperparameters that exist in the low-level training algorithm MAPPO. Hyperparameters are not specified by search but by directly using the recommended variables used by the low-level training algorithm MAPPO, except for the number of parallel environments which is limited by the core of our server. In SMAC environments, the number of neighborhood iterations is set to 8.

All results displayed in this paper are trained on servers with a 16-cores xeon-6130 2.10 GHz CPU with 64GB memory, and an NVIDIA V100 24GB GPU.

|  | Neighborhood size $m$ |
| --- | --- |
| 5mvs6m | 3 |
| mmm2 | 2 |
| 3s5zvs3z6z | 5 |
| 27mvs30m | 15 |
| 10mvs11m | 5 |

Table 5: Hyperparameter Table for neighborhood size used in RLNS and BLNS.

## B   Additional Experiment Results

We have included our training curve in Fig. 1 and Fig. 2. While it is acknowledgeable that the training curves have many fluctuations due to the reported values being median values, we can still see the same conclusion as we get from the table in the main paper: our proposed algorithms are as good as our base-level algorithm MAPPO. A reasonable neighborhood size $m$ that is larger than 3 can also make our algorithm not significantly worse than the base algorithm.

Besides, as shown in Fig. 1a, we observe that ALNS may not always be the most efficient at any time, given that training on a limited number of agents like 2 may lead to a slow improvement on policy function, but ALNS is as good as the low-level algorithm as the training progress and ALNS increase the neighborhood size.

## C   Additional Theoretical Guarantee for Convergence of MARL-LNS

In the main paper, we have provided the theorem that guarantees the convergence of MARL-LNS to be the same as the low-level algorithm. Here we provide a stronger theorem in the case that in each LNS iteration, the policy is updated to local optimal.

**Theorem 2** *(Adapted from Lyu & Li (2020)) Assume the expected cumulative reward function $\mathcal{J}$ is continuously differentiable with Lipschitz gradient and convex in each neighborhood partition, and the training by*

| Hyperparameters | value |
|---|---|
| recurrent data chunk length | 10 |
| gradient clip norm | 10.0 |
| gae lamda | 0.95 |
| gamma | 0.99 |
| value loss | huber loss |
| huber delta | 10.0 |
| batch size num | envs $\times$ buffer length $\times$ num agents |
| mini batch size | batch size / mini-batch |
| optimizer | Adam |
| optimizer epsilon | 1e-5 |
| weight decay | 0 |
| network initialization | Orthogonal |
| use reward normalization | True |
| use feature normalization | True |
| num envs (SMAC) | 8 |
| num envs (GRF) | 15 |
| buffer length | 400 |
| num GRU layers | 1 |
| RNN hidden state dim | 64 |
| fc layer dim | 64 |
| num fc | 2 |
| num fc after | 1 |

Table 6: Hyperparameter table for the MAPPO training part used in BLNS and RLNS.

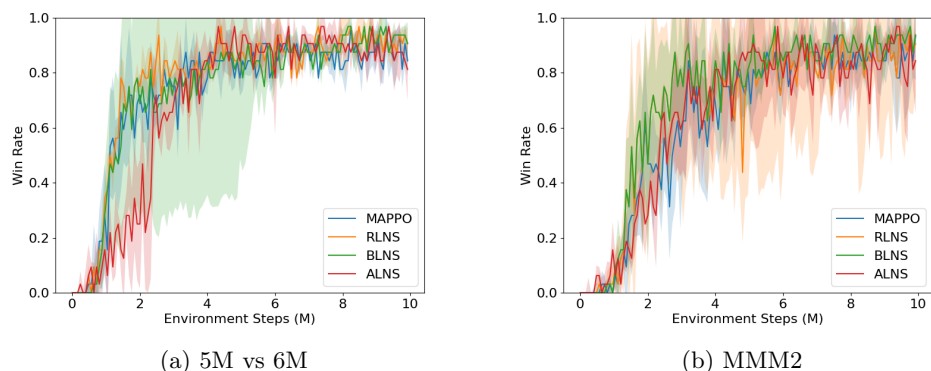

(a) 5M vs 6M            (b) MMM2

Figure 1: Median value and standard deviation of the RLNS, BLNS, and ALNS training curves compared to MAPPO on two SMAC scenarios. Although the neighborhood size is set as half of the total number of agents, the training curves are not much different.

*the low-level algorithm guarantees that the training happening on the i-th neighborhood is bounded by some high-dimension vector $w_i$. Suppose $\sum_{i=1}^{\infty} w_i^2 < \infty$, then the following hold:*

*1. If $\sum_{i=1}^{\infty} |w_i| = \infty$, then for any initial starting point of training, the training can converge to a stationary point.*

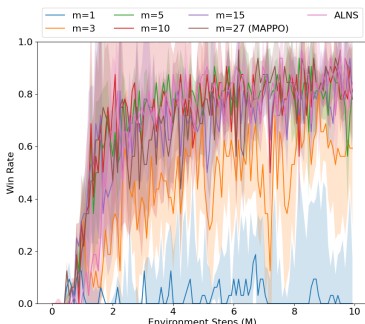 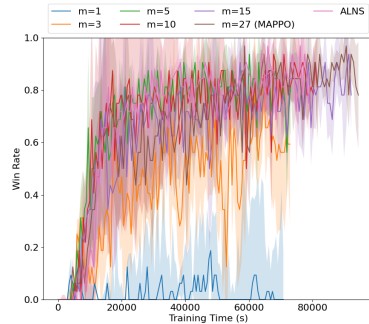

(a) Win rate corresponds vs. environment step  (b) Win rate corresponds vs. wall clock time

Figure 2: Median value and standard deviation of the BLNS training curve on the 27m_vs_30m scenario on SMAC for different neighborhood sizes $m$.

2. *If $\pi^k$ is optimized to optimal in each LNS iteration, then there exists some constant $c > 0$ such that for $i \geq 1$,*

$$min_{1 \leq k \leq i} \sup_{\pi_0 \in \Pi} [- \inf_{\pi \in \Pi} \langle \nabla \mathcal{J}_{\pi^k}, \frac{\pi - \pi^k}{|\pi - \pi^k|} \rangle] \leq \frac{c}{\sum_{k=1}^{i} w_k}$$

Compared to the one used in the main paper, this theorem relies on the assumption of the optimality gap to be summable. However in general practice, always making the policy optimized to optimal in each LNS iteration will lead to an extremely long training time and, thus, is less preferable.

