# OpenReview forum: "MARL-LNS: Efficient Multi-agent Deep Reinforcement Learning via Large Neighborhoods Search"
_TMLR — Rejected by TMLR_

### Review · Reviewer_Wv3j · 2025-03-17

**Summary Of Contributions:**

This paper introduces **MARL-LNS**, a novel framework for cooperative multi-agent reinforcement learning (MARL) that accelerates training by iteratively optimizing subsets of agents ("neighborhoods") using existing deep MARL algorithms. Key contributions include:

1. **Algorithmic Framework**: A generalizable training paradigm that reduces the curse of dimensionality by alternating training on agent subsets, requiring no additional trainable parameters.
2. **Variants**: Three practical instantiations—Random (RLNS), Batch (BLNS), and Adaptive (ALNS) Large Neighborhood Search—each differing in neighborhood selection strategies.
3. **Theoretical Guarantees**: A convergence analysis linking MARL-LNS to block coordinate descent, showing that optimal policies under the framework match those of the base algorithm.

**Audience:**

Yes

**Broader Impact Concerns:**

Experiments rely on high-end GPUs (V100/P100). Discuss implications for resource-constrained settings and potential optimizations for lower-tier hardware. How well does it perform in real-world scenarios, and how fast is its inference speed?

**Claims And Evidence:**

Yes

**Requested Changes:**

1. Include experiments or discussion on environments where full agent coordination is critical (e.g., MGS) to clarify boundary conditions.

2. Test MARL-LNS in scenarios with heterogeneous agents or prioritized roles (e.g., emergency vehicles in traffic control) to assess generality.

3. Analyze causes of RLNS instability (e.g., in 3s5zvs3s6z) and propose mitigations.

4. Compare with additional MARL acceleration methods (e.g., asynchronous frameworks) to contextualize time-saving claims.

**Strengths And Weaknesses:**

*Strengths*:

- **Practical Efficiency**: Addresses the CPU-GPU bottleneck in MARL training via data reduction, offering significant wall-clock time savings.
- **Theoretical Rigor**: Connects MARL-LNS to block coordinate descent, providing convergence guarantees under standard assumptions.
- **Comprehensive Evaluation**: Extensive experiments across diverse scenarios (e.g., SMAC super-hard maps) validate robustness and generalizability.

*Weaknesses*:

- **Limitations in Tightly-Coupled Environments**: Fails in tasks requiring full agent coordination (e.g., MGS), as noted in societal impact.
- **Assumption of Agent Homogeneity**: Relies on parameter sharing and uniform agent importance, limiting applicability to heterogeneous/priority-driven scenarios.
- **Variance in Results**: RLNS exhibits high variance in certain scenarios (e.g., 3s5zvs3s6z), suggesting instability in purely random neighborhood selection.

---

> ### Author Response · Authors · 2025-03-27
>
> Thank you for the review. Here we address your concerns one by one:
>
> 1. Indeed, as you noted, we have already acknowledged that our algorithm will not work in MGS. We recognize that the title of the subsection may have been misleading, so we have updated it to "Limitations" to clarify that we are discussing domains where our proposed method may not be effective. On the other hand, MGS not only requires all agents to collaborate, but also provides very little guidance when they do not, which is why our algorithm fails in this context. As discussed in our paper, many tasks can be divided into smaller subtasks to provide guidance, as demonstrated in the SMAC and GRF experiments, where our algorithm shows an advantage.
>
> 2. This is essentially the same point as in item 1, which has already been addressed in the paragraph. We have acknowledged that our proposed algorithm does not perform well in these cases. Since our approach introduces no additional hyperparameters to tune and provides a speedup without compromising policy quality, we believe it is reasonable for it to have some constraints on its applicable domains.
>
> 3. The instability in RLNS stems from its random neighborhood selection mechanism. This motivated our proposed variants - BLNS and ALNS - which address this limitation through more principled selection strategies. As demonstrated in Section 4.3, these approaches significantly improve stability while maintaining the computational benefits of the original framework.
>
> 4. While prior work on MARL acceleration is limited, we have discussed existing approaches in our related work section (final paragraph), including the asynchronous frameworks. Our method achieves comparable speedup ratios to these state-of-the-art techniques. However, we note that none of these prior approaches provide well-maintained implementations that would enable direct empirical comparison under identical experimental conditions.
>
> 5. (Broader Implications). While the speedup depends on the codebase and server configurations used, we want to emphasize that our speedup ratio is predictable, as shown in Eq. 1 of our paper. Our algorithm is particularly beneficial when the GPU-side computations are slower, and when training on the GPU constitutes a larger proportion of the total time used.

---

### Review · Reviewer_uVp2 · 2025-03-26

**Summary Of Contributions:**

The paper introduces MARL-LNS for improving the efficiency of cooperative MARL by LNS. The key contributions are:
- Framework Proposal: MARL-LNS iteratively trains subsets of agents (neighborhoods) using existing deep MARL algorithms as low-level trainers, reducing training time without additional trainable parameters.
- Algorithm Variants: Three variants are proposed—Random LNS (RLNS), Batch LNS (BLNS), and Adaptive LNS (ALNS)—each differing in how agent subsets are alternated during training.
- Theoretical Guarantees: The paper provides theoretical convergence guarantees, showing that MARL-LNS maintains the same convergence properties as the underlying MARL algorithm.
- Empirical Validation: Extensive experiments on StarCraft Multi-Agent Challenge (SMAC) and Google Research Football (GRF) environments demonstrate that MARL-LNS reduces training time by at least 10% while achieving comparable or superior performance to state-of-the-art methods.

**Audience:**

Yes

**Claims And Evidence:**

Yes

**Requested Changes:**

Addressing the weaknesses mentioned above is critical to securing my recommendation for acceptance.

Typos:
- k or $ \kappa $ in Eq (1)?
- Section2: "explored neighbor a sampling strategy"
- Table 1: "The 50% and 70% Total steps" -> total steps or Total Steps?

**Strengths And Weaknesses:**

Strengths:
- This paper is well-written and easy to follow.
- The authors extensively discuss related aspects of the propsed approach, including thorough abaltion and discussion on neighbourhood selection heuristics.
- The framework is agnostic to the underlying MARL algorithm, making it widely applicable.
- The convergence guarantees provide a theoretical rigor.


Weaknesses:
- A 10% reduction in training time does not seem to represent strong empirical performance. Given that the paper evaluates only two test environments, it is unclear whether the improvements generalize to other applications or are merely a result of environmental specificity and careful hyperparameter selection (the authors note, for instance, that the learning rate must be carefully tuned).
- While the authors claim the framework is agnostic to the low-level MARL algorithm, the paper only demonstrates integration with MAPPO, leaving its broader applicability unverified.
- Since training time depends heavily on CPU/GPU configurations, the methodology for ensuring a fair comparison requires further elaboration.
- The rationale behind baseline selection needs clarification. Given that the proposed algorithm is essentially an extension of iterative training in MARL, does it make sense to also compare against algorithms that rely on iterative training?

---

> ### Author Response · Authors · 2025-03-27
>
> Thank you for the response, and the detailed pointers to the typos. We have updated our draft to reflect them with changes highlighted in red. Here we answer to the concerns you raised in the weaknesses in order:
>
> 1. We believe you have raised multiple points within this comment, and we will address them individually. First, the time savings achieved by our approach are predictable, as shown in Eq. 1 of our paper. Our empirical results closely align with these theoretical predictions, further validating the consistency of the savings. Second, as discussed in the last paragraph of our related works section, several other studies have also explored accelerating MARL algorithms, and their reported speedups are on the same order as ours. Finally, while strong theoretical guarantees may depend on hyperparameter selection—as noted in Sec. 4.1—our experiments (as described in Sec. 5.1) used the default hyperparameters of the base algorithm (MAPPO) without additional tuning, underscoring the practicality of our method.
>
> 2. While our paper currently focuses on MAPPO as the underlying low-level learner, this choice is well-justified: MAPPO is the most widely adopted algorithm in MARL and offers stronger theoretical guarantees, as established in our analysis (Sec. 3.2). We acknowledge that our initial claim may have been overly broad, and we have revised the paper to clarify that our primary contribution is MAPPO-LNS, a method designed to accelerate MAPPO—consistent with prior work in this space (as discussed in the final paragraph of our related work section, existing optimizations also specifically target MAPPO). Extending our high-level framework to other base algorithms remains an exciting direction for future work.
>
> 3. To address your concern effectively, we would appreciate specific details regarding this point. We emphasize that the algorithm's design itself provides theoretical guarantees independent of empirical outcomes: predictable time savings (per Eq. 1) and non-decreasing performance. Our empirical evaluation was conducted specifically to demonstrate the robustness of these theoretical findings in practical MARL settings, confirming their applicability despite environmental uncertainties. Please let us know which specific theoretical or empirical aspect requires further discussion.
>
> 4. We agree that iterative training is an important ablation study. In fact, iterative training corresponds to the special case of $m=1$ in our ablation analysis. While this approach yields meaningful time savings per iteration, we observe that the resulting policy underperforms when evaluated at equivalent wall-clock time (see Appendix Fig. 2b). This finding is consistent with established practices, where iterative training is typically employed in competitive environments (e.g., the prisoner's dilemma or game-theoretic scenarios), rather than in cooperative environments like SMAC and football, which are the focus of our work.

---

### Review · Reviewer_MqQk · 2025-03-27

**Summary Of Contributions:**

In this paper the authors propose the application of large neighborhood search to multi-agent reinforcement learning. They discuss their framework, which takes the intuition of neighborhood search and adapts it to MARL in terms of retaining existing parts of the solution and only updating the current policy within the neighborhood. The authors propose three MARL-LNS algorithms and demonstrate substantial speedups in two environments with roughly comparable performance to an ablation with no large neighborhood search. The paper's contributions are the framework for applying LNS to MARL, the three simple algorithms, and the experimental results.

**Audience:**

No

**Broader Impact Concerns:**

No broader impact concerns except to the extent that papers with fabricated citations may be getting submitted to TLMR.

**Claims And Evidence:**

No

**Requested Changes:**

1. The authors should carefully review the paper for fabricated citations and other potentially hallucinated text. Any instances should be removed. Ideally I'd also like to see the authors give some explanation for how this happened in the first place, if they are allowed to resubmit.
2. The authors should cite the prior work combining LNS and MARL and ideally compare against it.
3. The authors should compare their approach to relevant baselines.
4. The authors should propose some algorithm that allows their framework to be applied to cases with heterogeneous task collaboration.

**Strengths And Weaknesses:**

The strength of the paper is in the application of LNS to MARL. It's a simple idea but it seems to work fairly well. The experiments are also relatively simple but the core of them is well-positioned technically: the authors demonstrate the approach in two domains, consider some ablations, and run on multiple seeds.

The current draft of the paper has a number of major weaknesses.

I normally wouldn't start with this, but the writing in the paper has serious issues. Most alarmingly, I found two invented citations in the paper. This suggests the possibility of generated text from an LLM being used for at least part of the paper writing process. The invented papers were

"Xunhan Hu, Jian Zhao, Wengang Zhou, and Houqiang Li. Discriminative experience replay for efficient
multi-agent reinforcement learning. CoRR, abs/2301.10574, 2023. doi: 10.48550/arXiv.2301.10574. URL
https://doi.org/10.48550/arXiv.2301.10574."
and
"Hanbaek Lyu. Convergence and complexity of block coordinate descent with diminishing radius for nonconvex optimization. arXiv preprint arXiv:2012.03503, 2020."

There are somewhat similar papers, "DIFFER: Decomposing Individual Reward for Fair Experience Replay in Multi-Agent Reinforcement Learning" for the former and "Convergence and complexity of block majorization-minimization for constrained block-riemannian optimization" for the latter, but this is still a major concern. I'll also note that I only checked the first half of citations before I found these. This throws the rest of the paper into serious question.

Outside of the fabricated citations, the language of the paper is poor and overly casual. The paper repeatedly uses "a lot", makes claims like "to get a huge overall reduction", and has unclear/confusing phrasing like "a number of
how many times of training is used". This makes the entire thing difficult to read and the authors' intended meaning unclear. But again, it's unclear to what extent the extent may have been generated and may represent hallucinations from an LLM.

Outside of these structural issues I have concerns with the paper's novelty and value. In terms of novelty, this is not the only application of LNS to MARL [1]. It's possible this work was happening in parallel given the timing but it should still be acknowledged since the paper was published last year. More broadly, LNS has already been applied to many domains, and the lack of technical complexity means that there is relatively little technical novelty in the paper broadly.

I also have concerns with the value of the paper. The authors present results and a discussion but despite a claim that "The time reduction is greater than that of other previous works on speeding up the overall time used in the training of MAPPO" there is no direct comparison to prior work. It is insufficient to compare speed-up times reported in prior work given the difference in domains and implementation. This claim would be much stronger if the authors actually compared to any baseline approaches.

1. Wang, Yutong, et al. "LNS2+ RL: Combining Multi-agent Reinforcement Learning with Large Neighborhood Search in Multi-agent Path Finding." arXiv preprint arXiv:2405.17794 (2024).

---

> ### Author Response · Authors · 2025-03-27
>
> Dear reviewer,
>
>
> Thank you for your detailed review. However, we respectfully disagree with the points you currently listed as requested changes.
>
>
> 1. We would like to respectfully point out that questioning the integrity of our work without thorough verification is a serious claim that warrants careful consideration. As indicated in your comments, both references cited were from ArXiv, a platform where updates are common. Specifically, the concerns you raised pertain to version 1 of the first paper and versions 1–3 of the second paper. To ensure accuracy, we have now updated all references to reflect their latest versions. We appreciate your diligence and hope this clarification addresses your concerns.
>
> 2. We acknowledge that the title of the cited "prior work" may suggest relevance to our paper and that its specialized domain knowledge may pose challenges for a thorough review. However, we would like to clarify that the cited work is specifically tailored to the MAPF problem and is not intended as a general framework applicable to broader MARL environments, nor is it directly comparable to our work. Nonetheless, we appreciate your suggestion and have added the paper as an additional reference in the context of using LNS in MAPF.
>
> 3. As you noted, the prior works mentioned in the last paragraph of our related work section are not directly comparable, as they are designed for specific implementations and environments. Furthermore, none of these works offer a well-maintained codebase that we could build upon. Therefore, while we have not included direct comparisons, we present the numbers to illustrate that a 10% reduction in time is a meaningful improvement.
>
> 4. We would like to emphasize that while our algorithm is primarily designed for homogeneous tasks, it also performs well in heterogeneous environments. In fact, the MMM2 and 3s5zvs3s6z scenarios in SMAC, as well as the GRF environment we tested, are heterogeneous. The limitation we discussed—using examples like the fire engine—refers specifically to the fact that our algorithm does not guarantee priority learning when agents are heterogeneous. However, since our approach introduces no additional hyperparameters to tune and provides a speedup without compromising policy quality, we believe it is reasonable for it to have some constraints on its applicable domains.

---

### Review · Reviewer_j5Ra · 2025-03-27

**Summary Of Contributions:**

The authors propose MARL-LNS, an approach where MARL agents learn to guide neighborhood search heuristics. The idea is well motivated and heuristic. Three MARL-LNS variants have been proposed. Experimentally, this work proves speedups while not sacrificing performance.

**Audience:**

Yes

**Claims And Evidence:**

Yes

**Requested Changes:**

See weakness above.

**Strengths And Weaknesses:**

Strengths:

1. The major contribution is the framework combining MARL and LNS. One advantage is that this framework applies to other MARL algorithms.
2. I appreciate the theoretical convergence guarantees in this work.

Weaknesses:

1. First, I have a concern regarding the novelty of this work. Can the authors comment on the substantial contribution of this work aside from the simple combination of MARL and LNS?
2. Experimentally, this work is limited to two test environments. The reduction in training time does not seem significant either. The generalizability of this work is uncertain.
3. I have concerns with the baseline selection in this work. Can the authors comment on why your selection is sufficient enough?

---

> ### Author Response · Authors · 2025-03-27
>
> Thank you for the time and effort you put into our paper. Here we answer the three weaknesses you identified one by one:
>
> 1. We believe that novelty is a highly subjective metric. Most importantly, to the best of our knowledge, no prior work has explored how to combine MARL and LNS. Even if the high-level idea involves integrating two existing approaches, many influential studies have followed a similar path and made significant contributions. For example, MAPPO can be seen as a combination of PPO and MADDPG, stable diffusion builds on principles from both image generation and the diffusion process used in physics, and vision transformers apply the transformer architecture to the vision domain. These works have meaningfully advanced their respective fields and received well-deserved recognition. In our case, combining MARL and LNS requires addressing key challenges, such as defining the optimization variables, determining the neighborhood selection policy, and deciding how to process selected neighborhoods. Traditional LNS approaches, which rely on destroying and rebuilding solutions, are infeasible for MARL due to the high cost of training from scratch. These considerations make our contribution both non-trivial and valuable.
>
> 2. We want to emphasize that the algorithm's design itself provides theoretical guarantees independent of empirical outcomes: predictable time savings (per Eq. 1) and non-decreasing performance. Our empirical evaluation was conducted specifically to demonstrate the robustness of these theoretical findings in practical MARL settings, confirming their applicability despite environmental uncertainties. Given the predictability of the MARL-LNS framework, we believe generalizability is not a major concern.
>
> 3. As you noted in the summary, our algorithm empirically demonstrates speedup without sacrificing performance. Regarding performance baselines, we use MAPPO, which serves as both our baseline and the low-level learner due to its widespread adoption and strong theoretical guarantees. Using MAPPO as the sole performance baseline directly supports our claim that our approach does not degrade performance. In terms of time reduction, we would have liked to compare our method to other speedup algorithms that achieve similar improvements, as discussed in the last paragraph of the related work section. However, none of these approaches provide a well-maintained codebase that would allow us to reimplement them under the exact same conditions. As a result, we instead present numbers to illustrate that a 10% speedup is a meaningful improvement for a speedup algorithm.

---

### Review · Reviewer_pvMP · 2025-05-05

**Summary Of Contributions:**

This paper proposed the MARL-LNS framework that could improve efficiency by iteratively training on alternating subsets of agents in low-level MAPPO learner. The authors introduce three algorithm variants RLNS, BLNS and ALNS to this framework. A theoretical convergence guarantee is provided and the empirical evaluations on SMAC and GRF supports this claim.

**Audience:**

Yes

**Claims And Evidence:**

Yes

**Requested Changes:**

1. See weaknesses mentioned above
2. Minor issue in P14 Appendix B Fig.??, missing figure number.

**Strengths And Weaknesses:**

Strengths
1. The paper is well written and easy to read.
2. Theoretical convergence guarantees are provided .
3. Well-constructed ablation study and thorough analysis.

Weaknesses
1. The evaluation only has 5 seeds, which may weaken the strong "speedup" claim made in this paper.
2. The convergence guarantee comes from MAPPO, is there an explanation on why the win rate of MARL-LNS is better than MAPPO in some of the tasks in Table 2?
3. The high variance in Table 2 and Figure 1(a) is not well explained. Why ALNS learns slower at the beginning stage in Figure 1(a)?
4. According to the paper, this framework could be extended to other MARL algorithms. Table 2 already uses IPPO and QMIX as baselines, is there any attempt to adapt this framework to those two algorithms? More baselines from relevant works would be appreciated.

---

> ### Author Response · Authors · 2025-05-06
> **Rebuttal**
>
> Dear reviewer,
>
>
> Thank you for your time on our paper. Here we address your concerns one by one:
> 1. We respectfully argue that using 5 random seeds is consistent with common practice in multi-agent reinforcement learning (MARL) research. For instance, MAPPO [1] employs 6 seeds, while SMACv2 [2] uses only 3. Furthermore, the primary purpose of our experiments is to validate that the proposed algorithms behave as expected both in terms of time efficiency and convergence behavior. Given this focus, we believe the current number of seeds is sufficient to support our claims and demonstrate the alignment between our theoretical predictions and empirical outcomes.
> 2. As noted in the paragraph preceding Section 4.2, satisfying the convergence guarantees requires adjusting certain hyperparameters—such as the learning rate—to ensure that the assumptions on $\omega$ hold. However, as clarified in Section 5.1,
> > To show that our algorithm does not introduce extra fine-tuning effort, we do not change the hyperparameters used in our low-level algorithm MAPPO, e.g., the learning rate, the batch size, etc, as well as the network designs, and environment configurations. We provide more details in the appendix. This will affect the conclusion of Thm. 1, but we will use our results to show that this is not affecting the effectiveness of our algorithm.
> 3. In our design, ALNS begins with a smaller neighborhood size ($m=2$), which reduces training time per iteration. This choice can lead to slower learning in more challenging scenarios, where a larger number of agents need to coordinate effectively. However, ALNS quickly catches up and eventually surpasses other configurations once the win rate exceeds 80%. Regarding the variance, we are unsure which specific instance the reviewer is referring to. Overall, the variance in our results is comparable to that of MAPPO. For example, in the scenario with the highest observed variance, 3s5zvs3s6z, MAPPO also shows a variance greater than 30, which is similar to that of ALNS. We would be glad to address this concern further if more details are provided.
> 4. As your review appears to have been submitted significantly later than those of the other reviewers, we are uncertain which version of the paper you received. If you are reviewing an earlier version that does not include our revisions, which can be identified by the absence of red text, we kindly invite you to consult the latest version. In that version, we have clarified that our current use of the MARL-LNS framework with MAPPO is primarily motivated by the popularity and widespread adoption of MAPPO. If you are already reviewing the updated version, we would like to emphasize that our focus on MAPPO as the underlying low-level learner is a deliberate and well-justified choice: MAPPO is the most widely adopted algorithm in MARL and offers stronger theoretical guarantees, as established in our analysis (Sec. 3.2). We acknowledge that our initial claim may have been overly broad, and we have revised the paper to clarify that our primary contribution is MAPPO-LNS, a method designed to accelerate MAPPO—consistent with prior work in this space (as discussed in the final paragraph of our related work section, existing optimizations also specifically target MAPPO). Extending our high-level framework to other base algorithms remains an exciting direction for future work.
> 5. (Requested Change 2) Thank you for bringing this to our attention. We have updated the paper accordingly to address the issue.

---

### Review · Reviewer_minQ · 2025-05-09

**Summary Of Contributions:**

This paper leverages neighborhood search to reduce the training time in multi-agent reinforcement learning (MARL) by iteratively training on alternating subsets of selected agents. It proposes three variants: Random LNS (RLNS), Batch LNS (BLNS), and Adaptive LNS (ALNS). Experiments on two MARL benchmarks demonstrate that the proposed methods can enhance the performance of the MAPPO backbone.

**Audience:**

Yes

**Claims And Evidence:**

No

**Requested Changes:**

Please see my comments and recommendations above.

**Strengths And Weaknesses:**

Pros

1. The paper demonstrates promising performance using a simple and intuitive idea.
1. The discussions in Section 6 are appreciated and offer valuable insights.

Cons
1. While the evaluation highlights the potential of the approach, the observed performance gains are marginal, and the robustness of the improvement remains unclear:
* The improvement is limited, and as seen in Figure 1, the convergence curves exhibit high variance. It is difficult to distinguish which algorithm consistently outperforms the others. A more rigorous statistical analysis is recommended to support the hypothesis that the proposed methods offer significant gains.
* The MAPPO baseline appears not to have been tuned for optimal performance. The paper uses shared hyperparameters, but such settings may not be optimal for MAPPO alone. It remains unclear whether a better-tuned MAPPO could match or exceed the performance of the proposed methods.
* The connection between empirical results and the theoretical analysis in Equation (1) is not clearly established.

2. Some claims are not fully substantiated:
* The claim that the approach “does not include any hand-crafted or learned heuristic to select the neighborhood” and that ALNS is “entirely hyperparameter-free” is misleading. In fact, the key hyperparameter $m$ exists and is set by heuristic rules, as shown in Algorithm 2.
* The claim that the method is generic and can enhance different MARL algorithms is not supported, as all experiments are conducted solely on MAPPO.

3. While I do not critique the novelty for being based on a simple idea, the paper lacks a thorough analysis of when and to what extent neighborhood search can meaningfully benefit the base algorithm. Although three variants are introduced, their performance appears mixed, and it remains unclear under what conditions one should be preferred over another. The claimed reduction in CPU/GPU time also seems device-dependent and lacks strong empirical support. I would encourage the authors to more clearly articulate the key conclusions and takeaways of this study, particularly regarding when and why neighborhood search can provide consistent benefits to MARL training.

---

### Decision · Action_Editor_a5ps · 2025-05-09

**Recommendation:** Reject

**Comment:**

1. Empirical evaluation is too narrow to substantiate the main claim.
  - Only two environments are considered; both are mid‑scale cooperative tasks. Reviewers uVp2 and j5Ra note that the claimed “algorithm‑agnostic, environment‑agnostic” benefits remain untested.
  - The sole baseline is vanilla MAPPO. There is no head‑to‑head comparison with other MARL speed‑up techniques (e.g., asynchronous MAPPO, locality‑aware sampling), despite the manuscript’s assertions of superior efficiency.
  - Hardware configuration and fairness of timing measurements are not adequately documented (uVp2).
2. Magnitude and practical relevance of the reported speed‑up.
3. All reviewers converge on the view that a 10 % reduction, while positive, is not compelling in isolation—especially given the added complexity of selecting neighborhoods and the lack of evidence that gains persist in larger‑scale or tightly‑coupled settings.
4. Methodological limitations not addressed experimentally.
5. Reviewer Wv3j highlights that MARL‑LNS assumes parameter sharing and homogeneous agents; it degrades on tasks requiring full‑team synchrony or heterogeneous roles, yet no experiments probe these edge cases.
6. Clarity and reproducibility issues.
7. Numerous typos, undefined symbols (e.g., k or  in Eq. 1), and casual phrasing impede comprehension (uVp2).
In summary, while the idea of borrowing LNS intuition for MARL is interesting, the manuscript suffers from serious technical and methodological weaknesses that must be rectified. Expanding and fairly benchmarking the empirical study, and clarifying the method’s applicability beyond the current narrow setting are essential next steps.

**Audience:**

Yes

**Claims And Evidence:**

The authors introduce MARL‑LNS, a large‑neighbourhood‑search (LNS) wrapper around existing cooperative multi‑agent RL algorithms (instantiated with MAPPO).

Claims: (i) iteratively training on alternating agent subsets yields at least a 10 % wall‑clock speed‑up with no loss in final performance; (ii) the framework is algorithm‑agnostic and enjoys the same convergence guarantees as the base learner.

Evidence: Empirical support is limited to two domains (SMAC and GRF) and one base algorithm. Reported gains are modest (≈10 %), and no direct comparisons are provided to other recently proposed acceleration methods or to iterative‑training baselines.

**Resubmission Of Major Revision:**

The authors may consider submitting a major revision at a later time.